# Expanding the Frontiers of Industrial Robots beyond Factories: Design and *in the Wild* Validation

Siméon Capy [1,*], Liz Rincon [1], Enrique Coronado [1,2], Shohei Hagane [1], Seiji Yamaguchi [1], Victor Leve [3], Yuichiro Kawasumi [3], Yasutoshi Kudou [3] and Gentiane Venture [1]

1. Department of Mechanical Systems Engineering, Tokyo University of Agriculture and Technology, Koganei, Tokyo 184-8588, Japan
2. Industrial Cyber-Physical Systems Research Center, National Institute of Advanced Industrial Science and Technology, Koto-ku, Tokyo 135-0064, Japan
3. Engineering Division, Kawada Robotics, Taito-ku, Tokyo 111-0036, Japan
* Correspondence: simeoncapy@gmail.com

**Abstract:** Robots able to coexist and interact with humans are key elements for Society 5.0. To produce the right expectations towards robots, it will be necessary to expose the true current capabilities of robots to the general public. In this context, Human–Robot Interaction (HRI) *in the wild* emerges as a relevant paradigm. In this article, we affront the challenge of bringing an industrial robot (NEXTAGE Open) outside factories and laboratories to be used in a public setting. We designed a multi-modal interactive scenario that integrates state-of-the-art sensory devices, deep learning methods for perception, and a human–machine graphical interface that monitors the system and provides useful information to participants. The main objective of the presented work is to build a robust and fully autonomous robotic system able to: (1) share the same space as humans, (2) work in a public and crowded space, and (3) provide an intuitive and engaging experience for a robotic exposition. In addition, we measured the attitudes, perceptions, expectations, and emotional reactions of volunteers. Results suggest that participants considered our proposed scenario as enjoyable, safe, interesting, and clear. Those points are also the main concerns of participants about sharing workspaces of daily environments with robots. However, we can point out some limitations with a biased population mainly composed of Japanese and males. In future work, we will improve our scenario with non-functional features or emotional expressions from the robot.

**Keywords:** human–robot interactions; robotic applications; robots in the wild; affecting computing

## 1. Introduction

Robots with advanced interactive capabilities are promising alternatives to more efficient and flexible industrial systems [1]. However, the lack of acceptance and positive attitudes towards robots among employees have limited the spread of collaborative robots in factories [2,3]. Moreover, it is expected that the current academic and commercial interest in the use of these advanced robots expands its frontiers beyond factories to more ecological and everyday-life scenarios, such as restaurants, shops, and homes [4]. Unlike traditional efforts in industrial robotics generally focused on increasing task performance, the next generation of robotics applications enabling interaction with humans require the consideration of hedonics (i.e., emotions and desires), as well as ergonomic factors [5–7]. These novel research activities will enable manufacturers and designers to build alternatives or counter-measures to increase the acceptability, the social impact, and the desirability of interactive robots [2,8]. Many research articles, such as [9–11], suggest that a low acceptance and negative attitudes toward robots are in part produced by the mismatch between people's expectations (in part due to social media, movies, or inexperience with robots) and the real capabilities of robots. Robots must be able to work outside laboratories to evaluate more realistic expectations. However, due to the complexity required to create robust

applications with robots, Human–Robot Interaction (HRI) applications are rarely tested in uncontrolled and public settings. Moreover, they are often presented and evaluated with convenient samples (e.g., laboratory and faculty members) [12,13]. This approach makes data collection more manageable and avoids many technical issues often presented when robots are required to perform in open, uncertain, and crowded environments. However, there is a need in the HRI community to move towards approaches that enable the acquisition of more valuable theoretical and technical insight through the use and validation of robotic systems in natural, open, everyday environments. This approach is called in literature as HRI *in the wild* [12,13]. Common types of robots used in this emergent paradigm are social and service robots. Some examples are presented in [14–16]. In many cases, these robots are either pre-programmed or remotely controlled rather than fully autonomous. Some recent examples using this methodology are [17–19]. In this article, we affronted the challenge of bringing an industrial robot *to the wild* through the development of a multi-modal and distributed system architecture. This architecture integrates advanced sensors, effective deep learning methods, and a human–machine graphical interface to enable fully autonomous HRI. This paper is organized as follows. Section 2 presents the related works. Section 3 presents the objectives and contributions of this article. Section 4 presents the proposed system architecture. Section 5 presents the experimental methodology. Section 6 presents the results. Discussion and conclusions follow.

## 2. Related work

Exposure to robots is basically done in three ways: *no HRI*, (i.e., participants are not asked to imagine, view, or interact with a robot), *indirect HRI*, (i.e., participants are asked to imagine an HRI task or observe an interaction with the robot using videos or images) and *direct HRI*, (i.e., the robot is physically present and interacts with participants) [3]. Several cross-cultural studies about acceptance and attitudes toward robots in different domains have been presented in previous works, such as [20–22]. However, in these types of studies, no HRI or indirect HRI is used. While the study of these non-functional and human-centered aspects through direct interaction is a popular topic in some social robotics areas, such as Robot-Assisted Therapy (RAT) [23], education [24] and elderly-care [25], the attention of these aspects when performing advanced Human–Robot Interaction (HRI) activities using industrial and collaborative robots is still limited [26]. A recent review of research articles evaluating attitudes, anxiety, acceptance, and trust in social robotics domains is presented in [3]. They discovered that "studies providing direct HRI may report different attitudes to studies where participants do not directly interact with a robot". Moreover, attitudes can change depending on the application domain and the design of the robot (e.g., humanoid-like or anthropomorphic). In the context of industrial robotics, studies collecting attitudes, expectations or perceptions towards robots after a direct HRI interaction are still rare. For example, Aaltonen et al. [27] grasped possible expectations of factory workers from the industrial and academic points of view. However, data collection was performed with an online questionnaire with no direct HRI. Moreover, experimental research with industrial and collaborative robots has been mainly limited to lab experiments with convenience samples (e.g., students or people working in a research lab) [28]. Laboratory studies are still mainstream in HRI. However, validations of robotics systems *in the wild* will increase more and more in relevance as the need to have advanced robotic systems able to be used outside factories and laboratories increases.

Museums and exhibitions are suitable *in the wild* scenarios for exposing novel techno-logical achievements to people with different backgrounds and interests [29]. An example is proposed in [30], where an ethnographic study is performed in a museum with a humanoid robot. Other examples of HRI demonstrations *in the wild* scenarios using FROG (a tour guide robot) are presented in [31,32]. Other suitable locations for *in the wild* experiment are shopping malls, such as the work done by Kand et al. [33]. They used a robot (Robovie-IIF) to help customers to find their way in the mall. The robot was partially teleoperated, to overcome speech recognition of algorithm difficulties, and to keep the interaction smooth.

We can also mention some of our previous work with robot interaction in school [14,15]. In those studies, the Pepper robot was used to entertain children in their school with some activities, in uncontrolled scenarios. In the current article, we affront the technical challenge of designing and executing an HRI application with a dual-arm industrial robot in a public and crowded scenario, specifically a robotic exhibition.

Table 1 summarizes recent and similar works reporting the development of robotic systems with industrial robots able to share the same space with people and that collect the users' perceptions towards robots after performing direct HRI. This table shows that one of this article's main novelties/differences against previous works is the experimental setting, which is not performed in a closed room or laboratory. Instead, validation of our proposed system is performed in a public and noisy scenario with no control of environmental conditions (e.g., illuminations or the number of people in the field of view of the robot). Similar to [26], our proposed system is integrated into a dual-arm industrial robot. Unlike [26], where the robot is remotely controlled, the robotic system proposed in this article is fully autonomous. Moreover, many systems developed in previous and similar works were developed to follow an industrial task, which in most cases requires some previous training. An exception is [34], where a straightforward but suitable task is proposed for enabling people with mental and physical disabilities to interact with an industrial robot. These complex industrial tasks, such as assembly, are inappropriate for a robotic exhibition where people interact with the robot voluntarily and have no time for complex explanations. Therefore, we designed an intuitive socio-emotional scenario where the robot can guide and adapt to human actions and emotions.

**Table 1.** Research articles proposing interactive robotic systems with industrial robots for grasping perceptions towards robots after performing direct Human–Robot Interaction.

| Article | Robotic Platform | Setting | Task | Autonomy | Training Required | Participants |
|---|---|---|---|---|---|---|
| Muller et al. [2] | Universal Robot 5 robot arm | Laboratory | Assembly task | Fully autonomous | Yes | 90 subjects mainly students from a technical university |
| Rossato et al. [35] | Universal Robot 10e robot arm | Laboratory | Collaborative task | Fully autonomous | Yes | 20 industrial senior and younger workers |
| Drolshagen et al. [34] | KUKA LBR iiwa 7 R800 (robot arm) | Closed room | The robot picks up wooden sticks to hand them over to the worker | Fully autonomous | No | 10 participants with mental or physical disabilities. |
| Elprama et al. [26] | Baxter dual-arm robot | Closed room | Participants instruct the robot to put blocks inside boxes | Remote controlled | Yes | 11 car factory employees |
| This work | NEXTAGE Open dual-arm robot | Public space | The robot gives gifts to visitors according to their instructions and facial expression | Fully autonomous | No | Hundreds, but only 207 answered some questionnaires. |

## 3. Objectives and Contributions

This article proposes a novel application where a dual-arm industrial robot, originally designed to be used by industrial workers, can interact with people from different backgrounds and outside laboratories and factories. The main objective of the project presented in this article is to develop an intuitive, social, and engaging interactive scenario for the International Robot EXhibition (IREX) using the NEXTAGE Open robot by *KAWADA ROBOTICS CORPORATION* [36]. Rather than proposing a typical scenario where robots need to be isolated from visitors or a cooperative industrial task (which often requires previous training), we developed an application where humans intuitively interact with an industrial robot through emotions and body motions. In this way, visitors can experience what it is like to interact with an advanced robot with advanced grasping, perceptual,

and cognitive capabilities in an everyday-like situation. Therefore, the main contributions of this work are (i) the creation of an intuitive and engaging HRI scenario that integrates the NEXTAGE Open dual-arm industrial robot; (ii) the execution and validation of the proposed HRI scenario in an unconstrained, crowded, and dynamic environment; and (iii) validation of the proposed scenario using self-reports that grasp the emotional experience towards the robot platform and the proposed HRI scenario of participants as well as their potential needs and attitudes towards industrial robots with social capabilities.

Moreover, this system is validated with more types of users, rather than only factory workers or faculty members (e.g., students).

## 4. Design and Implementation

### 4.1. Hardware

The experimental setup is composed of four parts. The main one is the *NEXTAGE Open* robot from Kawada Robotics [36]. It is an upper-body anthropomorphic robot with 15 DoFs, 6 for each arm, 2 in the neck and 1 for the waist. The robot also has 4 cameras, 2 in the head and 1 in each hand. The hands are composed of a three-fingered pneumatic gripper, and the payload for each is 1.5 kg. The robot is controlled by an Intel *NUC* PC on Ubuntu 16.

The second part is composed of three sensors used to detect the user's interactions with the robots. The **people detection** is achieved with two devices (redundancy for robustness): a *RealSense* depth camera (RS), and a set of ultrasonic sensors (US). The latter is a custom-made array of six *HC-SR04* sensors, see Figure 1. Both are used to estimate the proximity of the user to start and sustain the interaction and are placed below the robot on the front panel. Only the sensitive part is visible, thanks to two slits. The **user's choice detection** is done with a *Leap motion* placed on the booth's edge, in front of the object's window. It detects the choice between three positions: left (chocolate), center (pen) and right (eraser). A mark was placed on the floor to indicate the user's ideal position, which maximizes detection.

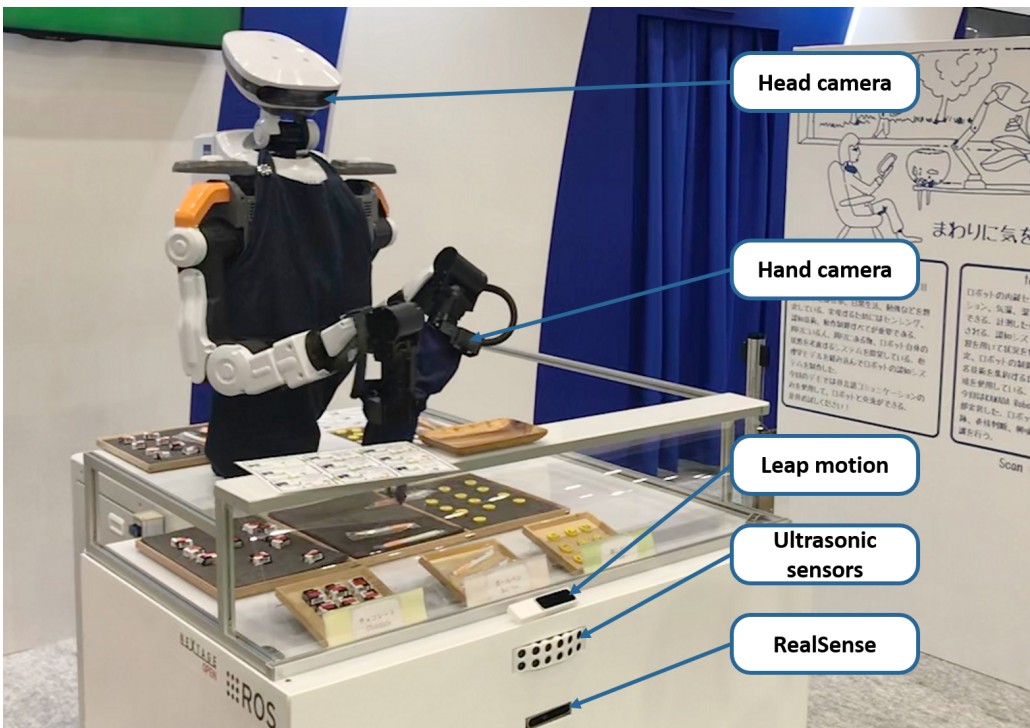

**Figure 1.** Robot on the booth.

The cameras are used for two purposes. The embedded **eye camera** (EC) detects the person's facial expression and the *engagement* of the user. According to those, the robot

will change its behavior, by knowing, for example, if the user is engaged in the interaction. The **hand cameras** (HC) are used to detect the position of the objects on the table, and to grasp them.

The third part is placed behind the robot: a monitor was installed on a wall to display the Graphic User Interface (GUI). It indicates to the user *what to do*, see Figure 2. The instructions are written in English and Japanese; visual feedback from the cameras (user's face with his engagement and the recognition of the objects from the HCs) is also displayed.

The last part consists of the computers. Besides the computer used to control the robot, two computers composed the apparatus. They run the algorithms that are divided on both to maximize robustness, see Section 4.2.

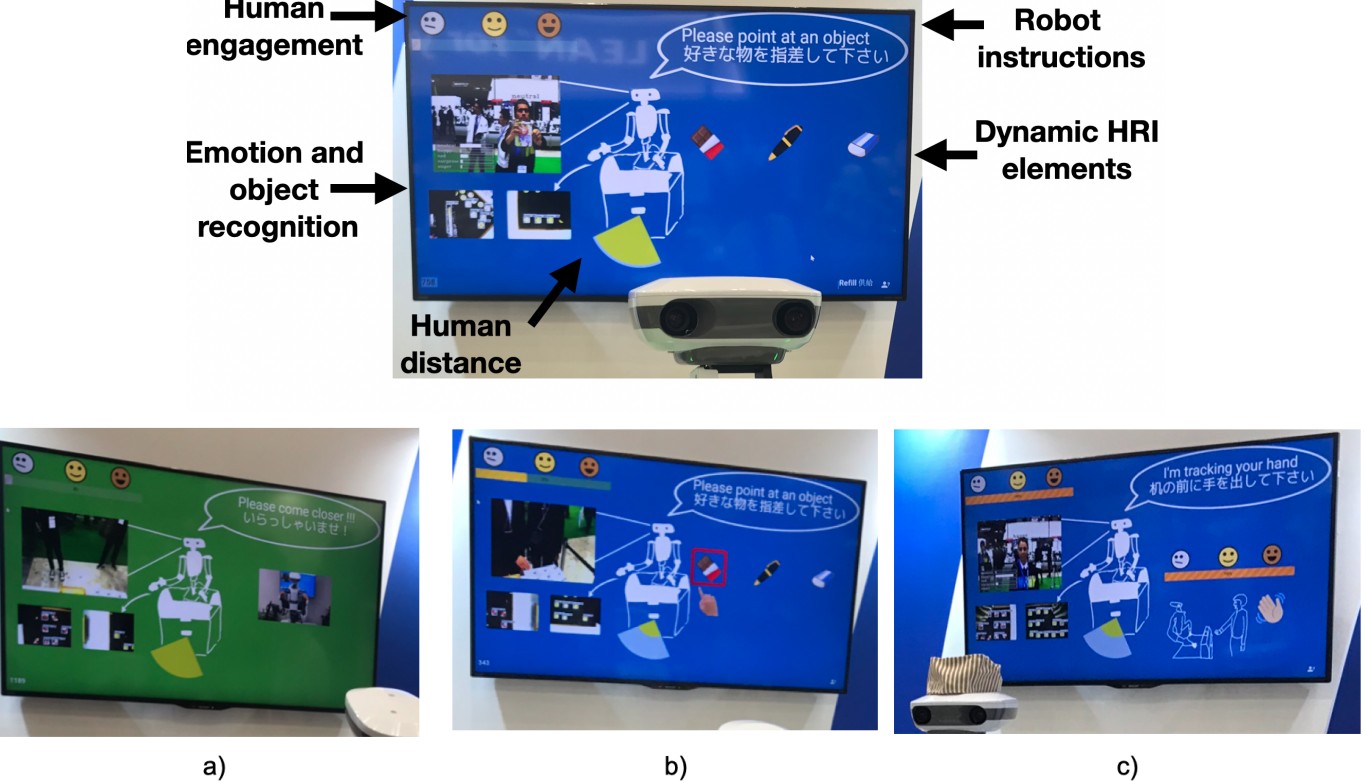

**Figure 2. Top**: main elements of the designed Graphical User Interface (GUI). **Bottom:** examples of elements shown in the GUI according to the status of the interaction: (**a**) No human interaction is performed, (**b**) the interface shows which object is selected by the human, (**c**) the robot provides information about what it is doing and how it feels.

### 4.2. Software

The software architecture is described in Figure 3. The first PC (Desktop: RAM 32 GB, CPU Ryzen 1900X 8 cores 3.8 GHz, GPU Nvidia GeForce RTX 2070) is dedicated to vision, with three resource-consuming algorithms. The second PC (laptop Dell Inspiron 14 5000: RAM 8 GB, CPU i7-7500 2.70 GHz) runs the sensors scripts. Data are gathered on two blackboard scripts (i.e., shared working memories) and sent to the control PC and the GUI using NEP [37], on a WLAN.

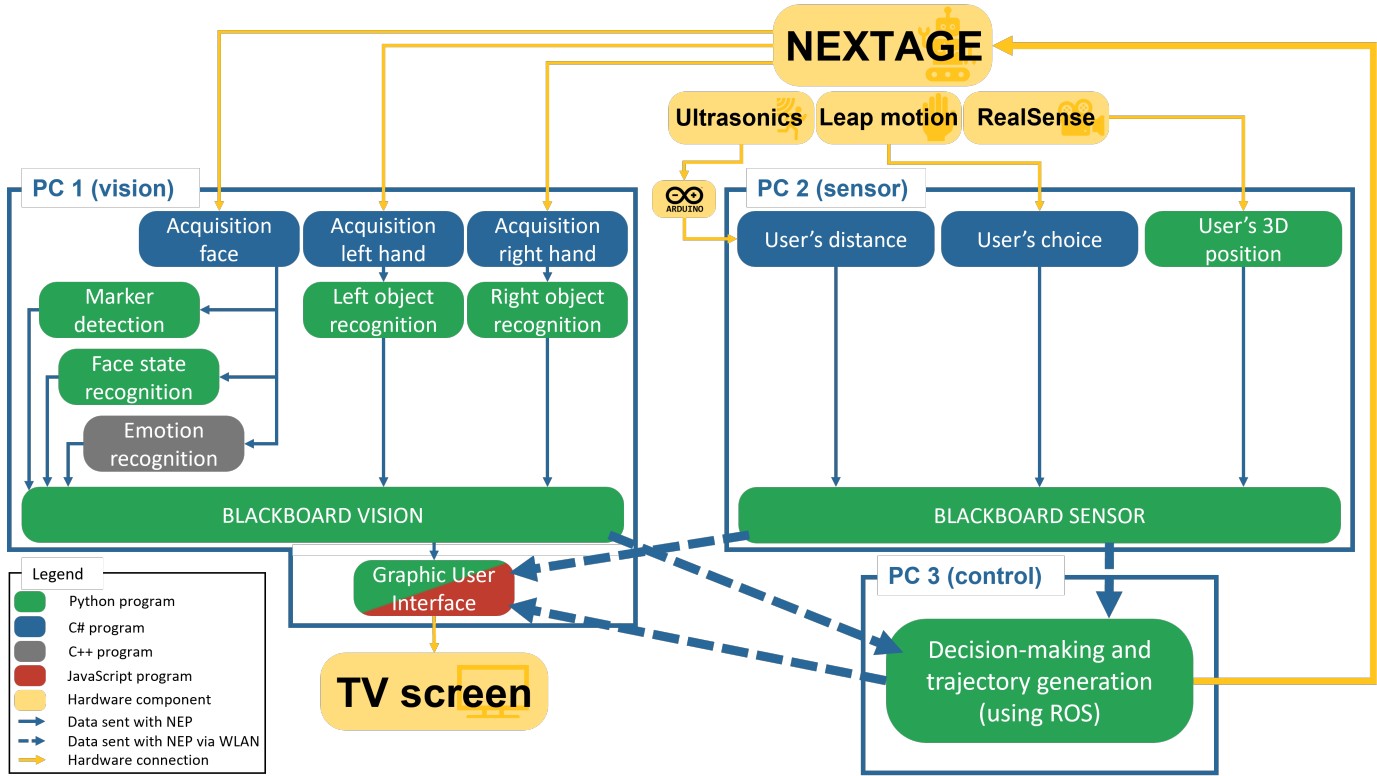

**Figure 3.** General architecture of the project. The GUI program is using client-server architecture, the server is in Python and the client in JavaScript.

### 4.2.1. Eye Camera Algorithms

The first algorithm **detects the gaze direction and head orientation** (and then determines the engagement) of the user regarding the robot. If s/he is not looking it in the eye, it will continue its task until the user is *engaged* in the interaction. We are using an algorithm we developed, described in [38].

The second one **recognizes five different emotions** of the user (We use the toolkit provided by OpenVINO: https://docs.openvino.ai/latest/omz_models_model_emotions_recognition_retail_0003.html-accessed on 2 December 2022), if s/he is *happy*, *sad*, *surprised*, *angry* or *neutral*. The emotions are used to determine if the robot will give the gift or not. If the emotion of the user is negative (e.g., *angry*) the robot will take the gift back and let the user choose another one. We assume the user changed their mind and would prefer another gift.

### 4.2.2. Hand Cameras Algorithms

Two identical algorithms run simultaneously to process the video stream of each hand. This algorithm performs object recognition by feature extraction and classification. Different customized models were created corresponding to the objects the robot has to manipulate. The Faster RCNN was applied, with these models taking advantage of its high capacity to detect different objects with enough precision.

The algorithm was trained to detect the three objects the robot has to grasp: a *pen*, a *chocolate* and an *eraser* (with a smiley shape) (Figure 4). The model was trained with about 300 images; with one or several objects, fully displayed or partially covered or with different luminosity and background.

The algorithm's output is the object's name, the bounding box (BB), and the confidence score (the probability of being a true positive). Only the objects detected with a confidence score over 75% are used, and the bounding box coordinates are sent to the control algorithm. Moreover, for the chocolate and the eraser, their BB is expected to be square, so the shape of the BB is checked. If it is a rectangle and not a square, it means the object is partially in

the field of vision. Hence, the center of the BB is not the center of the object, and to avoid grasping problems, the object is discarded until it becomes fully visible.

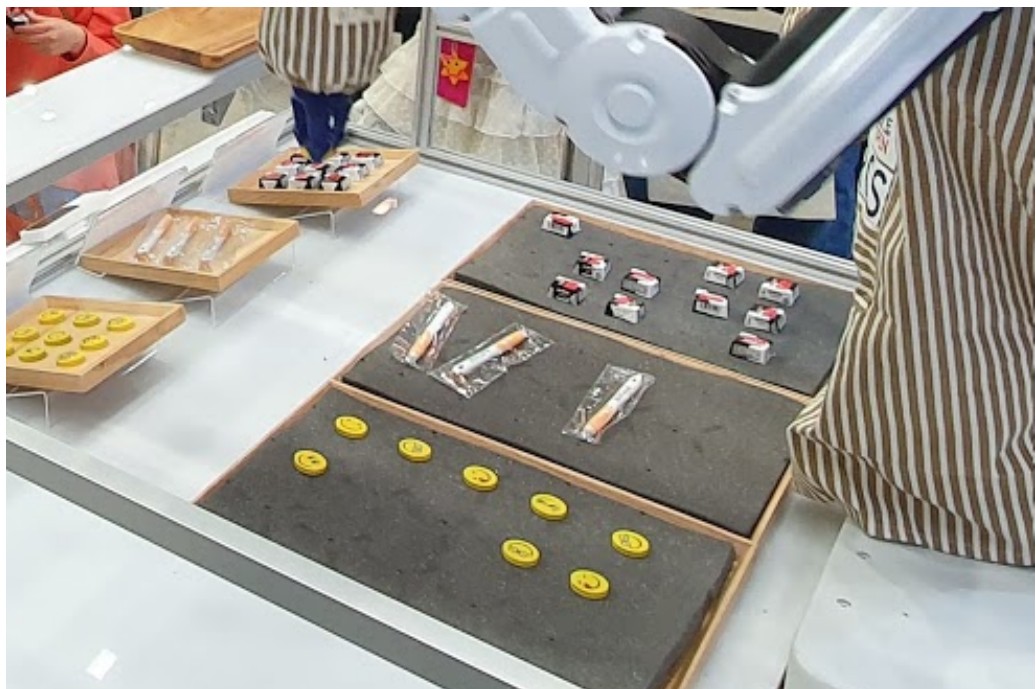

**Figure 4.** Real scene with the objects the robot has to detect and grasp.

### 4.2.3. Sensors Algorithms

The algorithm is separated into two parts. The first one is on an Arduino where the six ultrasonic sensors (US) are connected. The script loops on them to obtain the current distance data and send it via serial communication to the sensor PC. A second algorithm grasps the data from the six sensors and sends them by NEP to the blackboard sensor script that gathers all the sensor data before sending them to the control algorithm.

### 4.2.4. Control Algorithm

The algorithm is detailed in Figure 5. Four threads are running in the background to gather the data sent via NEP. Moreover, the algorithm is using a virtual grid to track the objects on the plates. The quantity and position of each object on that grid are recorded. The cameras are used to know the precise position of the different objects and facilitate the grasping task.

The **Working** task is used when no one interacts with the robot. In this case, the robot is moving objects on the grid randomly. When a user is detected, the robot stops this task and turns its head to the user. If it has an object in its hand, it proposes it to the user and gives it to him/her, if the facial expression and engagement are positive.

The robot then **Presents the objects** with a hand gesture (and on the TV screen as well) and waits for the user to point at the object of their choice, the pointing is detected by the Leap Motion sensor. The robot picks the selected object and offers it to the user. While doing that, the user's facial expression and engagement are checked, and according to them, the robot will give the object or pick a new one (see Section 4.2.1).

The **Refill task** is done when the robot has nothing to do with the user (i.e., during the **Working** task) if the plates are almost empty. If they are empty, the task is done even if a user is present. To refill, the robot turns 90° to access a big plate with different objects. This plate is easily accessible by the staff even during the robot's running, with safety. Because the objects are placed randomly on that plate, the robot relies only on the hand cameras with the object detection algorithm to pick the correct object.

For simplification, the **Invite next user** action is not in Figure 5. This action is done after two interactions with the same user. Indeed, we limited users to 2 consecutive interactions to avoid monopolization.

### 4.2.5. Human–Robot Graphical User Interface

We designed a Graphical User Interface (GUI) able to: (a) provide feedback to people interacting with the robot about the decisions and actions taken by the robot (e.g., the current facial expression of the user that has been detected, and selected gift), (b) give instructions to users about how to proceed in the interaction, and (c) enable developers to monitor the status of sensors and the results from deep learning algorithms. This GUI was developed using modern web technologies and JavaScript libraries, such as Node.js, HTML, CSS, and Vue.js. The interface subscribes to the blackboard sensor and blackboard vision modules and the decision-making modules using NEP. The information obtained from these external modules is used to dynamically change the elements displayed to the users. Figure 2 presents the basic sections of this interface and examples of the different elements changing according to the robot and sensors' status. When there is no interaction between the robot and a user, the background color of the interface is green; otherwise, it is blue.

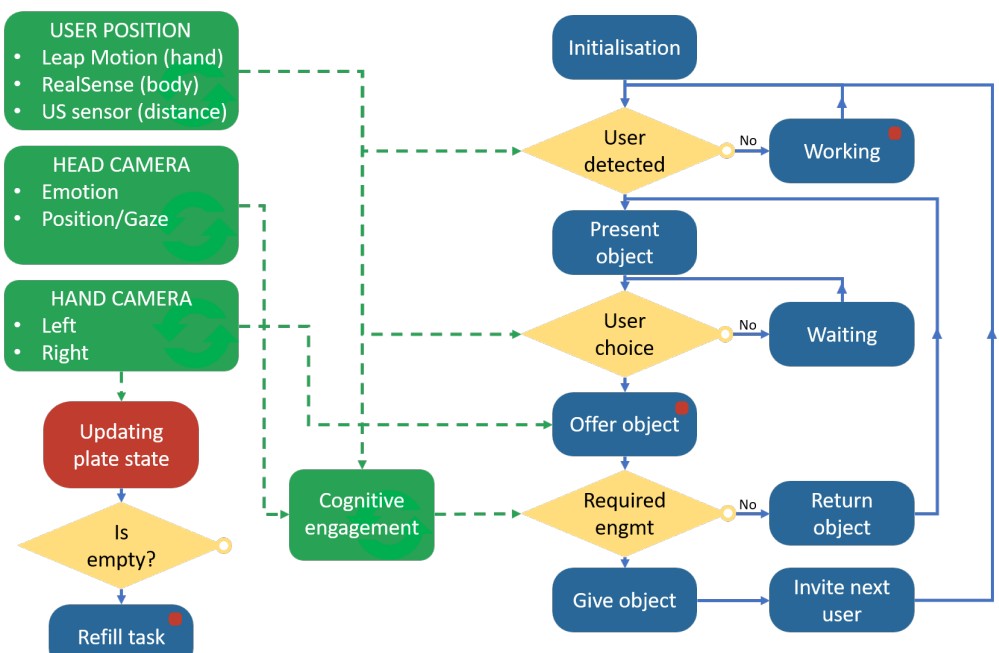

**Figure 5.** Flowchart of the command of the robot. The green boxes are threads running in the background and updating data. The actions with a red mark are calling the *Updating plate state* one. The *Cognitive engagement* thread estimates the engagement of the user towards the robot thanks to the user's position, emotion or gaze.

## 5. Validation *in the Wild*

The experimentation took place at the International Robotics Exhibition (IREX) 2019, in Tokyo, Japan. It is a public event for four days at the end of December, before the COVID-19 pandemic. As presented in Section 3, the main objective of this work was to build a system architecture that integrates state-of-the-art perceptual tools to provide engaging experiences for visitors of IREX 2019. In order to know if this objective was met and to identify possible improvements for posterior iterations, we formulated the research question Q1 (see below). Additionally to the main goal of this article, we also grasped the attitudes and expectations of visitors of IREX after a direct interaction with an industrial robot with affective and cognitive skills. Therefore, we formulated the next two additional research questions Q2 and Q3.

**Q1:** *What are the emotional reactions and perceptions of visitors towards the proposed interactive scenario?*

**Q2:** *What is the attitude of visitors towards robots after direct interactions with an industrial robot with affective and cognitive skills?*

**Q3:** *What are the potential expectations of visitors towards robots in their working and everyday environment?*

*5.1. Subjective Validation of the Proposed System*

We use semantic difference (SD) self-reporting questionnaires to grasp participants' emotional reactions to the robot and the proposed HRI. The results from these questionnaires are used to answer the research question Q1. In order to select the items of this questionnaire, we follow recommendations presented in the Kansei Engineering ergonomics discipline [39,40]. Kansei Engineering is a suitable human-centered and ergonomic approach often used to grasp and analyze the emotional, need, and social values of people towards products, interfaces, and services [41]. The first step in developing a self-reporting questionnaire is to collect a certain number of emotional words and adjectives that are relevant to the design of some specific product or application. Therefore, these adjectives can vary according to the application domain [39]. Additionally, these adjectives can be obtained after consulting experts in the domain or reviewing state-of-art articles. We collected 30 possible pairs from state-of-art works on industrial and social robotics. Finally, 18 pairs were selected to be applied in different 5-point Kansei (K) questionnaires. We classified these pairs of words into two sections: K-1 and K-2, which are presented in Tables 2 and 3, respectively. While K-1 was designed to grasp the feeling of visitors about the HRI scenario (using 6 pairs), K-2 was designed to grasp the impressions about robot design, usefulness, and skills (using 13 pairs). Additionally, two open questions, identified as OP-1 and OP-2, were applied to answer question Q3. These two questions were defined as follows:

**OP-1:** *If you had to work together with a robot, what would be the main characteristics you think the robot should have?*

**OP-2:** *If you had to live with a robot, what would be the main characteristics you think the robot should have?*

We use the highly cited Negative Attitudes toward Robots Scale (NARS) [42] questionnaire to grasp the attitudes of participants after a direct interaction with the robot and in this way answer question Q2. According to [3], NARS is the most popular questionnaire used in robotics to measure attitudes towards robots. This questionnaire comprises 14 questions with a 5-point Likert scale (1: I strongly disagree, 2: I disagree, 3: Undecided, 4: I agree, 5: I strongly agree). These questions are classified into three sections: NARS-S1, NARS-S2, and NARS-S3. Generally, the average value of these three sections is calculated and reported separately. These average values range between 1 to 5. The NARS-S1 section is composed of six questions and is designed to obtain attitudes toward situations of interactions with robots. The NARS-S2 section is composed of 5 questions and is designed to obtain attitudes toward the social influence of robots. Finally, the NARS-S3 section is composed of 3 questions and is designed to obtain attitudes toward emotions in interaction with robots [43]. Then, NARS-S1, NARS-S2, and NARS-S3 values are combined (average) to obtain the general NARS score, the value of which is also between 1 to 5. We use the original version of NARS defined in [42], which is in the Japanese language, and its version in English.

**Table 2.** Semantic analysis results (K-1), about the feelings the participant had regarding the robot. The *concepts* ($\mathcal{C}$) are regarding to the **satisfaction** ($\alpha$) and **comfort** ($\beta$).

| $\mathcal{C}$ | Dimension | | Semantic Evaluation ($\mu^{\sigma}$) | | | | | | | | | |
|---|---|---|---|---|---|---|---|---|---|---|---|---|
| | Positive (1) | Negative (5) | Japanese | Non-J. | *p* | Male | Female | *p* | Novice | Expert | *p* | Total |
| $\alpha$ | Happy | Unhappy | $1.43^{0.68}$ | $1.14^{0.38}$ | 0.136 | $1.46^{0.71}$ | $1.29^{0.56}$ | 0.346 | $1.24^{0.60}$ | $1.55^{0.67}$ | 0.109 | $1.38^{0.64}$ |
| $\alpha$ | Interested | Boring | $1.28^{0.55}$ | $1.14^{0.38}$ | 0.447 | $1.27^{0.53}$ | $1.24^{0.54}$ | 0.844 | $1.24^{0.52}$ | $1.27^{0.55}$ | 0.836 | $1.26^{0.52}$ |
| $\alpha$ | Disappointed | Amused | $4.60^{0.63}$ | $5.00^{0.00}$ | **0.000** | $4.54^{0.65}$ | $4.81^{0.51}$ | 0.116 | $0.72^{0.61}$ | $4.59^{0.59}$ | 0.467 | $4.66^{0.59}$ |
| $\beta$ | Relaxed | Anxious | $2.10^{1.22}$ | $1.86^{1.46}$ | 0.690 | $1.88^{0.99}$ | $2.29^{1.49}$ | 0.297 | $2.28^{1.43}$ | $1.82^{0.96}$ | 0.196 | $2.06^{1.23}$ |
| $\beta$ | Safe | Danger | $1.25^{0.54}$ | $1.71^{1.50}$ | 0.447 | $1.23^{0.51}$ | $1.43^{0.98}$ | 0.409 | $1.28^{0.89}$ | $1.36^{0.58}$ | 0.702 | $1.32^{0.75}$ |
| $\beta$ | Confused | Clear | $4.20^{1.14}$ | $3.71^{1.70}$ | 0.491 | $4.23^{1.18}$ | $4.00^{1.30}$ | 0.532 | $4.20^{1.29}$ | $4.05^{1.17}$ | 0.669 | $4.13^{1.21}$ |

**Table 3.** Semantic analysis results (K-2). The *concepts* ($\mathcal{C}$) are regarding to the **behavior** ($\gamma$), **interactions** ($\delta$) and **appearance** ($\varepsilon$).

| $\mathcal{C}$ | Dimension | | Semantic Evaluation ($\mu^{\sigma}$) | | | | | | | | | |
|---|---|---|---|---|---|---|---|---|---|---|---|---|
| | Positive (1) | Negative (5) | Japanese | Non-J. | *p* | Male | Female | *p* | Novice | Expert | *p* | Total |
| $\gamma$ | Smart | Stupid | $1.61^{0.72}$ | $1.33^{0.52}$ | 0.290 | $1.44^{0.58}$ | $1.76^{0.83}$ | 0.176 | $1.57^{0.66}$ | $1.57^{0.75}$ | 0.977 | $1.57^{0.69}$ |
| $\gamma$ | Simple | Complicated | $3.58^{1.00}$ | $3.17^{1.72}$ | 0.590 | $3.67^{1.11}$ | $3.29^{1.10}$ | 0.284 | $3.61^{1.12}$ | $3.43^{1.12}$ | 0.597 | $3.52^{1.10}$ |
| $\gamma$ | Dynamic | Static | $3.53^{1.20}$ | $2.20^{1.10}$ | **0.050** | $3.26^{1.10}$ | $3.56^{1.50}$ | 0.488 | $3.36^{1.33}$ | $3.38^{1.20}$ | 0.964 | $3.37^{1.24}$ |
| $\gamma$ | Responsive | Slow | $3.00^{1.23}$ | $2.50^{0.55}$ | 0.16 | $2.96^{1.26}$ | $2.88^{1.05}$ | 0.820 | $3.04^{1.26}$ | $2.81^{1.08}$ | 0.511 | $2.93^{1.16}$ |
| $\delta$ | Lifelike | Artificial | $2.89^{1.29}$ | $2.17^{1.17}$ | 0.205 | $2.48^{1.19}$ | $3.29^{1.31}$ | **0.046** | $2.57^{1.27}$ | $3.05^{1.28}$ | 0.218 | $2.80^{1.27}$ |
| $\delta$ | Emotional | Emotionless | $3.05^{1.11}$ | $2.83^{1.33}$ | 0.714 | $2.93^{1.11}$ | $3.18^{1.19}$ | 0.489 | $3.04^{1.15}$ | $3.00^{1.14}$ | 0.900 | $3.02^{1.12}$ |
| $\delta$ | Useful | Useless | $1.89^{1.01}$ | $1.50^{0.84}$ | 0.330 | $1.81^{0.96}$ | $1.88^{1.05}$ | 0.832 | $1.65^{0.88}$ | $2.05^{1.07}$ | 0.192 | $1.84^{0.98}$ |
| $\delta$ | Familiar | Unknown | $3.39^{1.35}$ | $2.50^{0.84}$ | 0.053 | $3.11^{1.45}$ | $3.53^{1.07}$ | 0.278 | $3.26^{1.39}$ | $3.29^{1.27}$ | 0.951 | $3.27^{1.30}$ |
| $\varepsilon$ | Desirable | Undesirable | $1.84^{0.89}$ | $1.20^{0.45}$ | **0.029** | $1.63^{0.84}$ | $2.00^{0.89}$ | 0.189 | $1.78^{1.04}$ | $1.75^{0.64}$ | 0.901 | $1.77^{0.86}$ |
| $\varepsilon$ | Cute | Ugly | $1.95^{1.09}$ | $1.33^{0.52}$ | **0.043** | $1.81^{0.96}$ | $1.94^{1.20}$ | 0.716 | $1.87^{0.97}$ | $1.86^{1.15}$ | 0.969 | $1.86^{1.04}$ |
| $\varepsilon$ | Modern | Old | $1.57^{0.83}$ | $1.33^{0.52}$ | 0.374 | $1.35^{0.69}$ | $1.82^{0.88}$ | 0.070 | $1.61^{0.78}$ | $1.45^{0.83}$ | 0.523 | $1.53^{0.79}$ |
| $\varepsilon$ | Attractive | Unattractive | $1.79^{1.04}$ | $1.33^{0.52}$ | 0.116 | $1.67^{1.00}$ | $1.82^{1.01}$ | 0.619 | $1.74^{1.01}$ | $1.71^{1.01}$ | 0.935 | $1.73^{0.99}$ |
| $\varepsilon$ | Like | Dislike | $1.68^{0.87}$ | $1.00^{0.00}$ | **0.000** | $1.44^{0.75}$ | $1.82^{0.95}$ | 0.175 | $1.65^{0.78}$ | $1.52^{0.93}$ | 0.623 | $1.59^{0.83}$ |

In total, NARS-S1, NARS-S2, and NARS-S3, and the proposed questions of K-1, K-2, OP-1, and OP-2, are composed of 35 questions. It is relevant to highlight that the research methodologies and objectives of this article are different from most classical HRI research activities performed in laboratories or structured environments, where participants are hired or are members of the same laboratory or school. This type of participant has time and motivation to effectively answer large questionnaires built with five or more items for the same psychological or usability construct. This traditional approach is often applied to prove the validity and reliability of the results using tools such as Cronbach's alpha [44]. In the application presented in this article, qualitative data is obtained from visitors of a robotic exhibition that participated voluntarily. Therefore, many typical considerations for increased validity, sensitivity, and reliability done in structured, descriptive, or explanatory research performed in laboratories are not suitable and are out of this project's scope. In this context, a common topic of discussion in the literature is when to use single or multi-items for the same construct in self-reporting questionnaires. For example, [45] discusses how the use of multiple-item measures is costly, aggravates respondent behaviour, and increases response errors. Moreover, even "a second or third item of the same construct contributes little to the information obtained from the first item". Diamantopoulos et al. in [46] suggest that single-item approaches are viable options in exploratory research. This type of research is usually performed at a preliminary stage, in unstructured settings, such as in this work. Recent works discussing the suitability of single-item or multi-item constructs are [47,48]. In practice, single-item approaches can be considered suitable for well-understood constructs, such as those measuring satisfaction. Moreover, multi-item approaches are more suitable for complex constructs, such as trust and attitudes [46,49]. In this work, we use a single-item approach for measuring satisfaction-related constructs and a multi-item approach (NARS) to measure attitudes. Martinez et al. describe in [50] that questionnaires for visitors in museums and expositions must be breve and simple, otherwise "it is unlikely that visitors will fill out the questionnaires if they are too long to complete or too difficult to understand, and responses will not reflect the real experience". Moreover, many practitioners, textbooks, and research articles suggest that long questionnaires should be avoided [51–53] to prevent careless responses and respondent fatigue as well as to motivate visitors to participate in the survey. Therefore, we divide the proposed 35 questions among four questionnaires (P1, P2, P3, and P4) that were applied on different days in the IREX exposition. NARS-S1 questions are asked in P1. NARS-S2 and NARS-S3 questions are asked in P2. K-1 (6 items), OP-1, and OP-2 questions are asked in P3. Finally, K-2 (13 items) questions are asked in P4. The first part of the questionnaire P4 (composed of 6 items) was used to grasp impressions of perceived intelligence and animacy towards the proposed robotics system. The second part of the questionnaire P4 (composed of 7 items) was used to grasp design-related aspects of the robot platform. Four demographic questions (age range, gender, country, and robotics experience) are included in P1, P2, P3, and P4. We provided two versions of each questionnaire, one in Japanese and the other in English, and participants were free to choose the language that suited them the most. Figure 6 sums up the content of each questionnaire.

### 5.2. Participants

The participants of this study are visitors of IREX 2019 that voluntarily interacted with the proposed HRI system. After they interacted with the robot, we asked them if they could fill out one of the questionnaires described before and by doing so give consent to use their answers for research purposes. No personal data were collected. The total number of participants answering is 207. From the questionnaires P1 and P2, participants (5 and 2, respectively) were discarded because they did not answer all of the questions. We discarded participants only in P1 and P2 because the NARS score is built from the answer to every question. Contrastingly, the answers to the questions of P3 and P4 can be taken individually, then, empty answers are not discarded. Table 4 sums up the demographic data of each questionnaire. Because IREX is held in Tokyo, Japan, most of the participants

are Japanese (between 70% and 86% depending on the questionnaires). We also divided the participants into groups according to their knowledge of robotics: *novice* (1/5 and 2/5 on the Likert scale), and people that are knowledgeable about robots, called *expert* (3/5, 4/5 and 5/5).

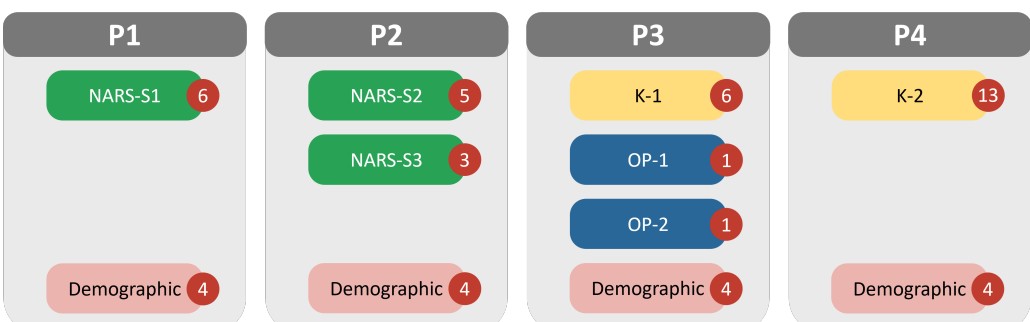

**Figure 6.** The content of each questionnaire P1–P4. The number in the red circle indicates the number of questions.

**Table 4.** Demographic data of each questionnaire. Some participants did not mention their gender or country. For the latter, we assume they were Japanese or not depending on the version of the questionnaire they used (Japanese or English).

|  | P1 | | P2 | | P3 | | P4 | |
|---|---|---|---|---|---|---|---|---|
| Considered answers | 78 | 100% | 67 | 100% | 47 | 100% | 44 | 100% |
| Japanese | 57 | 73% | 47 | 70% | 40 | 85% | 38 | 86% |
| Non Japanese | 22 | 28% | 20 | 30% | 7 | 15% | 6 | 14% |
| Male | 62 | 79% | 55 | 82% | 26 | 55% | 27 | 61% |
| Female | 15 | 19% | 12 | 18% | 21 | 45% | 17 | 39% |
| Novice | 21 | 27% | 22 | 33% | 25 | 53% | 23 | 52% |
| Expert | 57 | 73% | 45 | 67% | 22 | 47% | 20 | 46% |
| OP-1 | | | | | 18 | 38% | | |
| OP-2 | | | | | 21 | 45% | | |

## 6. Results

### 6.1. User Perceptions and Emotional Reactions

We use the results from questionnaires P3 and P4 to evaluate the answer to Q1 (*What are the emotional reactions and perceptions of visitors towards the proposed interactive scenario?*). Results from emotional reactions (P3) are shown in Table 2. Table 3 contains the result from the P4 questionnaire, regarding the feeling of the participants regarding the robot. We can gather some dimensions into similar *concepts* ($\mathcal{C}$), regarding the **satisfaction** ($\alpha$) and **comfort** ($\beta$) of the participants after interaction with the robot (K-1); and **behavior** ($\gamma$), **interactions** ($\delta$) and **appearance** ($\varepsilon$) for their impressions (K-2).

### 6.2. Negative Attitudes Towards Robots

We use the results from questionnaires P1 and P2 to answer research question Q2 (*What is the attitude of visitors towards robots after direct interactions with an industrial robot with affective and cognitive skills?*). The mean value and standard deviation (SD) for NARS-S1, NARS-S2 and NARS-S3 are summarized in Table 5. The mean values of each NARS section can be interpreted as positive (values close to 1), neutral (values close to 3), and negative (values close to 5). We divided the participants into *male–female*, *novice–expert* and *Japanese–non-Japanese* groups. Table 6 shows the mean and standard deviation values for each of these groups.

**Table 5.** Average ($\mu$) and standard deviation ($\sigma$) values for each NARS type. Values close to one represent positive attitudes and values close to five represent negative attitudes. Because the NARS-S3 is using positive questions, it is the five's complement of the mean that is shown.

| Type | $\mu$ | $\sigma$ |
|---|---|---|
| Interaction (S1) | 1.90 | 0.72 |
| Social (S2) | 2.60 | 0.96 |
| Emotion (S3) | 2.62 | 0.97 |

**Table 6.** Average ($\mu$) and standard deviation ($\sigma$) of the NARS questionnaire according to each different group. The column $p$ represents the p-value of Welch's $t$-test. Because the NARS-S3 is using positive questions, it is the five's complement of the mean that is shown.

| Groups | Interaction (S1) | | | Social (S2) | | | Emotion (S3) | | |
|---|---|---|---|---|---|---|---|---|---|
| | $\mu$ | $\sigma$ | $p$ | $\mu$ | $\sigma$ | $p$ | $\bar{\mu}$ | $\sigma$ | $p$ |
| Japanese | 1.90 | 0.64 | 0.890 | 2.47 | 0.97 | 0.084 | 2.73 | 1.01 | 0.130 |
| Non Japanese | 1.87 | 0.92 | | 2.90 | 0.88 | | 2.37 | 0.82 | |
| Male | 1.91 | 0.71 | 0.890 | 2.61 | 1.02 | 0.809 | 2.62 | 0.99 | 0.963 |
| Female | 1.88 | 0.78 | | 2.55 | 0.66 | | 2.61 | 0.86 | |
| Novice | 2.07 | 0.65 | 0.168 | 2.71 | 0.85 | 0.484 | 2.59 | 0.92 | 0.852 |
| Expert | 1.83 | 0.74 | | 2.54 | 1.01 | | 2.64 | 1.00 | |

### 6.3. User Needs and Desires

We use the open OP-1, and OP-2 to grasp the potential needs and design desires of users towards robots. Relevant words used by participants to describe needs and desired features of robots in working environments are: *kindness*, *safety*, *intuitive*, *responsive*, *cuteness*, *convenient*, *fast*, *accurate* and *efficient*. On the other hand, participants consider that robots in their every-life environment must be *fun*, *kind*, *safe* and *cute*. Moreover, they consider that robots must be able to have effective and interesting communication skills, understand feelings and emotions, use clothes, and have convenient and interesting functionalities such as being able to cook and dance.

### 6.4. Hypothesis

With the collected data and the different groups created, we can draw some hypotheses. They will lead, in addition to the research question, the analysis of the results.

**H1.** *The cultural background (country of origin) has an influence on their perception of the robot.*

**H2.** *The gender of the participants has an influence on their perception of the robot.*

**H3.** *The knowledge about the robots has an influence on their perception of the robot.*

Regarding the cultural background (H1), Jiang and Cheng pointed out better acceptance of robots by Chinese people [54]; and Bröhl et al. mentioned that Japanese people are more used to seeing robots in everyday life than Chinese or US people [55].

Concerning the knowledge about robots (H3), we can hypothesize that *expert* people have a more rational preconception about robots, on their real capability and not based on a fictional image of robots [56].

## 7. Discussion

### 7.1. Regarding Q1 and Q2

In Table 2, the visitors expressed having a highly positive experience by feeling **happy** ($\mu = 1.38; \sigma = 0.64$), **relaxed** ($\mu = 2.06; \sigma = 1.23$), and **interested** ($\mu = 1.26; \sigma = 0.52$) in the proposed HRI scenario. Moreover, rather than considering interaction with the NEXTAGE Open industrial robot a dangerous task, visitors mostly expressed feeling **safe** ($\mu = 1.32; \sigma = 0.75$). They also considered the proposed HRI scenario as **clear** ($\bar{\mu} = 0.87; \sigma = 1.21$; in order to keep coherence, the values in the text are presented to have

the value close to 1 for the described adjective. When an adjective is close to the *5-side*, then the *five's complement* $(6 - \mu)$ is used, and symbolized by a bar: $\bar{\mu}$) or intuitive enough, with a value really close to 1, which was the main objectives of this project. We can sum up the feeling of the participants and reply to Q1 by analyzing the different concepts, thus they were **satisfied** [$\alpha$] (scores around 1.3) and felt **comfortable** [$\beta$] (scores between 1.3 and 2) after their interaction with the robot.

Results shown in Table 3 suggest that visitors perceived the robot as a **smart** ($\mu = 1.67$; $\sigma = 0.67$) and **complex** machine ($\mu = 3.52$; $\sigma = 1.10$), which is between a **lifelike** and an **artificial** entity ($\mu = 2.80$; $\sigma = 1.27$). Even though the robot was able to recognize emotions as well as make decisions regarding them, or the engagement, visitors had a neutral impression between **emotional—emotionless** ($\mu = 3.02$; $\sigma = 1.12$). One factor influencing this result may be the lack of whole-body expressive movements presented in the proposed application, which could be one possible improvement in the system to have a better response from the participants [57]. This lack of emotion could also be an explanation for the perception of the robot's movement by the participants. They considered them neither responsive nor slow as well as mostly static rather than dynamic, with a score of around 3 for both of them. It is confirmed by the analysis of their impressions regarding the **behaviour** [$\gamma$] and **interactions** [$\delta$] of the robot; participants have difficulties interpreting them, and grade them neutrally with scores of around 3, aside from the intelligence and usefulness, which are judged positively.

From Table 2, we can see that the perception of the robot is positive, the visitors see it as **cute** ($\mu = 1.86$; $\sigma = 1.04$), **desirable** ($\mu = 1.77$; $\sigma = 0.86$) or **attractive** ($\mu = 1.73$; $\sigma = 0.99$). During the experiment, the robot was dressed in a hat and an apron, which could also explain the positive reaction of the public. However, they were quite neutral regarding their **familiarity** ($\mu = 3.27$; $\sigma = 1.30$) with NEXTAGE, even if it is not famous to the general public, its design is still close to what a person could expect to be a robot, and the participants judged its **appearance** [$\varepsilon$] positively (scores around 1.7).

Nevertheless, the analysis of the different subgroups cannot allow us to answer the three hypotheses (H1–3). Indeed, almost all the p-values of Welch's *t*-test do not permit us to discard the null hypothesis. This is clearer with the *novice–expert* group, where most of the p-values are close to 1. We can still point out some significant differences, with $p < 0.05$. We can observe some differences in the perception of the robot, especially regarding the country of origin. Thus, the Japanese participants described the presented robot as less **dynamic** ($\mu = 3.53$; $\sigma = 1.20$) than the foreign participants ($\mu = 2.20$; $\sigma = 1.10$). One explanation could be the habituation of robots in everyday life in Japan. Those robots, such as Pepper, try to be dynamic to catch the audience's attention. Yet, NEXTAGE is originally an industrial robot and has more *rigid* movements. We can see with our results that foreigners ($\mu = 2.50$; $\sigma = 0.84$) seem more **familiar** with that kind of robot than Japanese participants ($\mu = 3.39$; $\sigma = 1.35$). However, both groups found the robot **cute** and **liked** it in a similar proportion. On the other hand, men tend to see more **life** ($\mu = 2.48$; $\sigma = 1.19$) in the robot than women ($\mu = 3.29$; $\sigma = 1.31$).

To answer the second research question (Q2), we can also use the NARS analysis of P1 and P2. As shown in Table 5, participants have a positive attitude towards interacting with industrial robots, a neutral and slightly positive attitude towards the social influence of robots, as well as a neutral attitude toward emotions in interaction with robots. Results from the Welch's *t*-test shown in Table 6 suggest that there is not a statistically significant effect between genders (*p*-value higher than 0.05), again dismissing H2. This result differs from those reported in [2], which suggested that women have more negative attitudes towards robots than men. However, our results agree with those recently reported in [3], which performed a systematic mapping of research articles exploring attitudes, trust, and acceptance in different contexts of social robotics. They found out that "the gender of the participants [is] not associated with their affective attitudes toward social robots". Similarly, the *t*-test applied to *novice* and *expert* groups indicates there are no statistically significant effects regarding robotic-related experiences (H3). A hypothesis could be that

experts in robotics can be people working in marketing for robotic companies, or workers in factories where robots and humans work isolated. Therefore, it may also be the first time they actually share the same space with an industrial robot in an interactive scenario for most of them. This can explain why results from people with more experience in robotics presented similar values to those identified as novices (in many cases families and tourists). On the other hand, the novice group, even though they know less about robots, can still be interested in them, and that is why they visited IREX. The analysis of the countries is also non-significant and does not allow us to verify H1. Even if [55] pointed out some differences in the perceptions of robots by countries. Even if some analysis using NARS already pointed out that the country of origin has an influence regarding robot [58].

As explained in Section 2, studies reporting results of attitudes, acceptance and trust towards robots change depending on the application domain, the type of exposure, and the design of the robot. Romeo and Lado found positive perceptions about robots during the pandemic in Spain for Generation Z (1997–2012) [59]. However, those results could just be a confirmation of the general tendency regarding the acceptance of robots in Spain or by the young generation. Therefore, robotics systems such as those that we presented in this article shall become more relevant in the near future. This study mainly showed positive attitudes and impressions of visitors towards the use of robots. This suggests that the main objective of the project presented in this work was met, proving that people with no experience could interact with a robot without prior teaching, in an enjoyable scenario. However, these impressions can be biased by the fact that humans received a gift. How this factor and other complex factors (such as the cultural background) may or not result in positive attitudes, acceptance or trust and how they affect the robot's perception of different attributes such as its behavior or appearance is out of the scope of this article and can be discussed in future iterations/studies of the proposed cognitive system.

### 7.2. Regarding Q3

Concerning the third research question (Q3, *What are the potential expectations of visitors towards robots in their working and everyday environment?*), the words and desired features used for replying to the OP-1 and OP-2 suggest that providing robots with pleasing design/aesthetics, intuitive communication skills, and enjoyable HRI activities can be relevant aspects to improving the user experience and desirability of robots in both working and every-life settings. This contrasts with the traditional utilitarian objectives of robotics, which focus on performance (i.e, efficiency and effectiveness), and this agrees with recent studies in HCI and HRI showing the importance of hedonic factors (e.g., aesthetics, pleasure, and emotions) for the successful integration of novel technological [39,40,60,61]. The identification of the needs and desires of possible users of robots is a relevant task for developing robots and applications with greater acceptance, social impact, and market penetration [62,63].

The participants also focus a lot on safety, especially in work settings, 39% of answers to OP-1 mention it. For instance, one participant mentioned that "*a human should be able to turn it off*". This comment can be influenced by science fiction books or films. Liang and Lee pointed out that people with exposure to that kind of media have higher fear of robots and AI [56]. They also identified that only 30% of the US population has no fear of robots and AIs, and actually most of them make no difference between fear of robots and AIs. This fear of robots/AIs can explain the sake of safety, it was one of the conditions of their acceptance in factories from the 60s [64].

Even if we mention before that participants are not solely looking for efficiency, it is still a recurrent comment, especially in the case of working with robots. The robots are essential for the so-called *Industry 5.0* to improve the efficiency of the process [65]. The people are expecting them to do their tasks "*fastly*" and "*accurately*".

### 7.3. Limitations

Because of the nature of the experiment, without any selection of participants, we have some bias in the population. Then, besides the questionnaire P4, all of the others have an unbalanced distribution, with around 80% of males for P1 and P2, and around 60% for P3. Even if the "I" of IREX stands for *international*, the number of non-Japanese is below 30%. Due to the lack of foreign people, we gather all the non-Japanese together, even if some disparities exist, even between close countries, such as France and Germany [58]. Being a heterogeneous group could explain the difficulty to interpret the results.

The difference in the size of the different subgroups can be an explanation for the non-statistical difference in the results, and why we cannot clearly answer the hypothesis. Since it was not the main goal of our experiment, it is not a problem, but can be taken into account for future works.

## 8. Conclusions and Future Work

Industrial robots are advanced machines able to generate engaging HRI scenarios that are difficult to reach with most social robots, such as those tasks requiring advanced manipulation of objects. However, most of them are evaluated and used in factories or research laboratories. In this work, we proposed the initial iteration of an advanced industrial robotic system able to deal with a noisy, crowded, and public environment (an international robotic exhibition). We successfully affronted the challenge of bringing this robot to the wild and made a step ahead toward developing systems helping in the understanding of those social and technical factors influencing the adoption of robots in everyday environments. The proposed system could perform during the whole event (four days) without any technical issues and could manage the interactions with hundreds of visitors. The system can recognize human faces, states, actions, and emotions from the user. Then, in an unscripted scenario, the robot could adapt its behaviors according to the user's. We demonstrated by using an industrial robot in this explorative study that those robots could be used with the general public, in the wild, without prior specific training. In addition, the results from applied questionnaires suggest that visitors considered our proposed scenario enjoyable, safe, and interesting, suggesting that the proposed work's main objective was met. The collected expectations of this preliminary study also coincide with previous studies showing the importance of non-functional elements, such as aesthetics, in HRI and HCI. Some participants also mentioned that safety and convenience were important points for them if they have to work with robots in the future. These remarks are particularly important as design guidelines as the participants that joined our experiment were mainly familiar with robots. These characteristics may be more prominent for the widespread adoption of robots by the general public. Non-verbal communication with the robot could be one direction to investigate to improve safety and convenience. Future studies will be focused on exploring how non-functional features, emotional expression, and different robot personalities, as well as the different errors presented in the interaction, can influence or result in an improvement in the attitudes, positive experiences, trust, and acceptance toward robots in everyday and public scenarios. In addition to checking with a wider population about the impact of gender, knowledge about robots and country of origin. Concerning the latter, it could be better to have a group for each nationality.

**Author Contributions:** Conceptualization, L.R., E.C. and G.V.; methodology, S.C., L.R., E.C., S.H. and S.Y.; software, S.C., L.R., E.C., S.H. and S.Y.; validation, G.V.; formal analysis, S.C.; funding acquisition, G.V.; investigation, S.C., L.R., E.C., S.H., S.Y. and G.V.; resources, V.L., Y.K. (Yuichiro Kawasumi) and Y.K. (Yasutoshi Kudou); data curation, S.C.; writing—original draft preparation, S.C.; writing—review and editing, V.L., Y.K. (Yuichiro Kawasumi), Y.K. (Yasutoshi Kudou) and G.V.; visualization, S.C.; supervision, G.V.; project administration, G.V. All authors have read and agreed to the published version of the manuscript.

**Funding:** This study was funded by KAWADA ROBOTICS.

**Institutional Review Board Statement:** The study followed the guidelines of the Ethics Committee of the Tokyo University of Agriculture and Technology, Tokyo, Japan.

**Informed Consent Statement:** By filling out the questionnaires, the participants were fully informed of the anonymity of the questionnaire and agreed.

**Data Availability Statement:** Not applicable.

**Conflicts of Interest:** The authors declare no conflict of interest.

## Abbreviations

The following abbreviations are used in this manuscript:

| | |
|---|---|
| BB | Bounding Box |
| CPU | Central Processing Unit |
| DoF | Degree of Fredom |
| EC | Eye Camera |
| GPU | Graphics Processing Unit |
| GUI | Graphical User Interface |
| HC | Hand Camera |
| HCI | Human–Computer Interaction |
| HRI | Human–Robot interaction |
| IREX | International Robotics EXhibition |
| OP | OPen question |
| P1–4 | Questionnaire 1–4 |
| RAT | Robot Assisted Therapy |
| RCNN | Region-based Convolutional Neural Network |
| RS | RealSense |
| US | Ultrasonic Sensor |
| WLAN | Wireless Local Area Network |

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
