# Peer review of "Expanding the Frontiers of Industrial Robots beyond Factories: Design and in the Wild Validation"

_machines, doi:10.3390/machines10121179_

Round 1

Reviewer 1 Report

The paper is well-organized and novel. There are a few typing errors. Please correct. 

Author Response

The authors thank the reviewer for their comments to improve the quality of our manuscript. We did a spell check on the paper to correct the mistakes.

Reviewer 2 Report

Machines-2024340 with the title: "Expanding the Frontiers of Industrial Robots Beyond Factories: Design and Validation in theWild of a Multimodal, Affective and Cognitive Robotic System." This paper addresses the challenge of taking an industrial robot (NEXTAGE Open) out of factories and laboratories to be used in a public environment. It designs a multimodal interactive scenario that integrates state-of-the-art sensory devices, deep learning methods for perception, and a graphical human-machine interface that monitors the system and provides useful information to the participants. The main goal of the presented work is to build a robust and fully autonomous robotic system capable of 1) sharing the same space as humans, 2) working in a crowded, public space, and 3) providing an intuitive and engaging experience for a robotic exhibit. In addition, we measure the attitudes, perceptions, expectations, and emotional reactions of the volunteers. The results suggest that participants considered a proposed scenario as pleasant, safe, and interesting.

To improve their work, I suggest.

1.       Reducing the title is too long

2.       Includes in the abstract section the most relevant results and a paragraph with the limitations of this research and future work

3.       Check the English grammar and spelling of the entire paper

4.       In Table 1 apply the table format in accordance with the MDPI format.

5.       Improve the resolution and quality of figure 5

6.        Reduce the number of figures and use the most relevant ones. The other figures can be placed in a dataset.

7.       To make your research more supportive, place the study data in a dataset; you can create one at https://data.mendeley.com/.

8.       Reinforce the conclusions section.

9.       Fixing the style of references applied to MDPI.

10.   Apply the journal's citation and referencing style.

Author Response

The authors thank the reviewer for their comments to improve the quality of our manuscript. We addressed the reviewer's comments as follows:

To improve their work, I suggest.

  1.       Reducing the title is too long

⇒ The title has been shortened

  1.       Includes in the abstract section the most relevant results and a paragraph with the limitations of this research and future work

⇒ We completed the abstract as suggested.

  1.       Check the English grammar and spelling of the entire paper

⇒ We did a spell check on the paper to correct the mistakes.

  1.       In Table 1 apply the table format in accordance with the MDPI format.

⇒ The table has been changed to MDPI’s layout.

  1.       Improve the resolution and quality of figure 5

⇒ The figure has been enlarged

  1.       Reduce the number of figures and use the most relevant ones. The other figures can be placed in a dataset.

⇒ We removed one figure and merged two others.

  1.       To make your research more supportive, place the study data in a dataset; you can create one at https://data.mendeley.com/.

⇒ Due to ethics policies, we cannot share our dataset.

  1.       Reinforce the conclusions section.

⇒ The conclusion has been reinforced. We hope it answers the reviewer's comment.

  1.       Fixing the style of references applied to MDPI.
  2.   Apply the journal's citation and referencing style.

⇒ We are using the LaTeX template provided by MDPI, hence the references should follow the guidelines. However, if the Editor board has some requests we can correct them accordingly.

Reviewer 3 Report

The paper presents an experimental study regarding design and validation in the wild of a multimodal , affective and Cognitive Robotic system.

The paper is well written and structured, and the authors explained every aspect of the paper.  I didn’t find or maybe I missed the description of identifying human emotion of the visitor and in this case maybe the authors should better highlight this aspect.

Best regards.

Author Response

The authors thank the reviewer for their comments to improve the quality of our manuscript. We added a footnote with the link to the documentation of the algorithm we used to detect the user’s emotions. We hope it answers the reviewer's comment.